# Three-Dimensional Finite Element Modeling of Drilling-Induced Damage in S2/FM94 Glass-Fiber-Reinforced Polymers (GFRPs)

**DOI:** 10.3390/ma15207052

**Published:** 2022-10-11

**Authors:** Shahryar Manzoor, Israr Ud Din, Khaled Giasin, Uğur Köklü, Kamran A. Khan, Stéphane Panier

**Affiliations:** 1Department of Materials Science and Mechanical Engineering, Ecole des Mines de Saint-Étienne, 42100 Saint-Étienne, France; 2Department of Aerospace Engineering, Khalifa University of Science and Technology (KUST), Abu Dhabi 127788, United Arab Emirates; 3School of Interdisciplinary Engineering and Sciences (SINES), National University of Sciences and Technology (NUST), Islamabad 44000, Pakistan; 4School of Mechanical and Design Engineering, University of Portsmouth, Portsmouth PO1 3DJ, UK; 5Department of Mechanical Engineering, Faculty of Engineering, Karamanoglu Mehmetbey University, Karaman 70100, Turkey; 6Laboratoire des Technologies Innovantes, LTI-EA 3899, Université de Picardie Jules Verne, 80025 Amiens, France

**Keywords:** GFRPs, drilling, damage modeling, meso-scale, FEM

## Abstract

Considering that the machining of composites particularly fiber-reinforced polymer composites (FRPCs) has remained a challenge associated with their heterogeneity and anisotropic nature, damage caused by drilling operations can be considerably mitigated by following optimum cutting parameters. In this work, we numerically evaluated the effects of cutting parameters, such as feed rate and spindle speed, on the thrust force and torque during the drilling of glass-fiber-reinforced polymers (GFRPs). A meso-scale, also known as unidirectional ply-level-based finite element modeling, was employed assuming an individual homogenized lamina with transversely isotropic material principal directions. To initiate the meso-scale damage in each lamina, 3D formulations of Hashin’s failure theory were used for fiber damage and Puck’s failure theory was implemented for matrix damage onset via user subroutine VUMAT in ABAQUS. The developed model accounted for the complex kinematics taking place at the drill–workpiece interface and accurately predicted the thrust force and torque profiles as compared with the experimental results. The thrust forces for various drilling parameters were predicted with a maximum of 10% error as compared with the experimental results. It was found that a combination of lower feed rates and higher spindle speeds reduced the thrust force, which in turn minimized the drilling-induced damage, thus providing useful guidelines for drilling operations with higher-quality products. Finally, the effect of coefficient of friction was also investigated. Accordingly, a higher coefficient of friction between the workpiece and drill-bit reduced the thrust force.

## 1. Introduction

Fiber-reinforced polymer composites (FRPCs) have become indispensable materials for lightweight aerospace structures capable of being embedded with functionalized materials and high energy absorption [1,2]. Although composites are manufactured near to net shape, hole drilling is inevitable for the subsequent assembling and joining of complex components [3,4]. In aeronautical structures, drilling for fasteners is common to assemble complex structures and a large number of holes are usually drilled in a single commercial aircraft. High material anisotropy due to the difference in the mechanical properties of the matrix and fiber makes drilling a challenging task for damage-free FRPC-based systems [5,6]. Indeed, FRPCs are prone to different damage modes, including fiber pullout, matrix cracking and delamination, as a consequence of drilling operations. Damage-free holes in composites are of utmost importance to achieve high joint strength and the subsequent high fatigue service life. 

In the aerospace industry, it has been reported that the rate of rejection due to drilling-induced delamination was as high as 60% [7]. The high rejection rates are expensive both in terms of cost and time since drilling is performed in the final stages of the assembly. Therefore, to enhance the throughput by reducing the rejection rate and increasing the efficiency of the drilling process, researchers have focused on the underlying damage mechanisms when drilling FRPCs [6,8,9]. The main aim behind all these studies was to understand and predict the damage evolution and interaction phenomena between the cutting tool and the FRPC-based laminates.

Lee et al. [6] performed experimental studies to find a correspondence between the cutting force and the feed rate with respect to material thickness and the diameter of the drill bit. Shyha et al. [10] studied the influence of fiber orientation and feed rate on the material removal performance during drilling. Similarly, Tsao et al. [11] completed experimental work to find the effect of diameter ratio, feed rate, and spindle speed on drilling-induced delamination for various step-core drills. The study of Giasin et al. [5] is based on the drilling of hybrid composite materials to study the effects of cutting parameters, environmentally friendly cooling technologies and fiber orientations on the resulting cutting forces and hole quality. Su et al. [12] proposed a novel approach to reduce delamination and the corresponding burr formation by utilizing the controlled step scheme. The findings of Mohan et al. [13] showed that the workpiece thickness, feed rate and cutting speed were the most important factors contributing to delamination. 

Similar to the challenges involved in the experimental set-up, the numerical modeling of the drilling process of FRPCs is also a difficult task attributed to the multiaxial stress state in addition to the nonlinearities occurring due to the damage and frictional contact. Initial modeling efforts, such as those of [8,9,14], are based on analytical modeling to estimate delamination using a linear elastic fracture mechanics (LEFMs) method to solve the critical thrust force that is responsible for delamination in FRPCs. Different special drill-bits were also studied to obtain delamination-free holes in these preliminary studies. However, such modeling is based on simplified assumptions, including isotropic assumption of the anisotropic materials. In addition, the underlying mechanics cannot be well understood with such analytical methods. 

To cater for the complexities and simulate the drilling process as similar to the experimental process as possible without compromising the important intricacies involved, numerical approaches, such as finite element modeling (FEM), are indispensable at present. For instance, the FEM numerical technique was efficiently used by Andreucci et al. [15] for the drilling of screws in a bone implant, which is a complex problem. Employing the virtual crack extension technique, the authors of [16,17,18] used a 2D approximation of the drilling process. They idealized the drilling process as a simplified orthogonal cutting process. Nevertheless, these models showed a low accuracy since they did not represent the complete 3D topology of the drill bit and the kinematics of the drilling process. 

Zitoune et al. [19,20] modeled delamination in FRPCs utilizing cohesive elements that are based on the continuum damage mechanics (CDM) approach, also used in [21] for delamination modeling in GFRPs. However, the drilling process in [19,20] was treated as a static problem without considering the effect of mass and inertia. Furthermore, the drill was modeled as a simple cone-shaped indenter rather than the actual drill bit. Likewise, Durão et al. [22,23] developed a simplified drilling model in which the drill acted as a punch with a rotary motion that pierced FRPCs. The results of delamination were found to be in good agreement with the analytical studies. Singh et al. [24] modeled the drilling process of unidirectional GFRP laminates. They postulated that the rotation of the drill-bit possessed minimum effects on the thrust force, and hence the rotation was not included in the numerical analysis. Moreover, a pre-drilled hole of 2 mm was made in the laminate to avoid the effects of the chisel edge. Phadnis et al. [1] performed a detailed experimental and numerical analysis to investigate the effects of drilling parameters on cutting forces during the drilling of carbon-fiber-reinforced polymer composites (CFRPs). Delamination was included in the modeling using the cohesive elements available in the ABAQUS library [25].

Isbilir et al. [26] performed a comprehensive numerical analysis using the commercial FEM-based software ABAQUS to predict the progressive damage taking place in each ply (intra-laminar damage) as a result of drilling forces employing 3D Hashin’s failure theory in CFRPs. As far as delamination onset and progression are concerned, cohesive elements following the bilinear traction-separation constitute law was embedded between the layers. The study also reported a comparison between step and conventional drills. A more recent research work by Hale et al. [27] discussed a macro-scale-based 3D FEM modeling for the optimization of the drilling of CFRPs compared to other experiments. In comparison to other studies reported in the literature, the work covered a greater number of parameters. These parameters not only involved the drilling parameters (cutting parameters), but also some numerical parameters including mass scaling, bulk viscosity, friction, strain rate and cohesive surface modeling. 

Previous studies, for instance, those in [3,27], offered more detailed and hence computationally expensive models (cohesive zone modeling). The present modeling approach is simple and yet the prediction capability is in good agreement with the experimental findings. The cutting forces and drilling-induced damage were predicted with reasonable accuracy when drilling S2/FM94 GFRPs. A user subroutine (VUMAT) was incorporated to model the damage initiation and the subsequent failure of fibers and matrix in GFRPs. The cutting force results from the FEM model were then compared with experimental results from our previous study [5] to appraise the accuracy of the developed modeling approach. The effects of the coefficient of friction were emphasized, which requires the careful selection of the two materials (workpiece and drill-bit) involved during the drilling. 

## 2. Experimental Parameters

The experimental results that were used for model validation were adopted from the research work of Giasin et al. [5]. Readers are referred to the aforementioned reference for further details about the machining set-up. However, a summary is presented for the sake of completeness. 

The material used in the drilling study was S2/FM94 GFRPs with a total of 54 layers embedded, having a stacking sequence of [0/90]_27s_. The overall dimensions of the specimen were maintained as 240 × 240 × 7.18 mm^3^. In addition, the drill-bit utilized had a 6 mm diameter made of TiAlN-coated carbide material, having a point angle and helix angle of 140° and 30°, respectively. The drill had a shank length of 28 mm, whereas the flute length was 75 mm. The specific choice of the drill bit for the mentioned material was made based on the previous literature [28,29,30]. Moreover, the hole size in the aerospace industry generally lies in the 4.8–10 mm range [28]. Spindle speed and feed rate values were also adopted from the previously referred literature. These works suggested the use of a feed rate in the range of 0.05–0.3 mm/rev for drilling GFRPs and similarly spindle speed depends on the allowable speed for drill-bit and can be in the range of 1000–9000 rpm. A total of 27 holes were drilled in the dry test conditions. It is worth mentioning that each test set produced nine holes with a fresh drill bit used each time to exclude the effect of tool wear on hole quality. Apart from that, all the experimentally reported values represent the mean values of the three test runs. The drilling parameters used in the numerical modeling are summarized in Table 1. 

## 3. Finite Element Modeling of the Drilling Process

FEM-based modeling is used with liberties to develop numerical models that can reasonably describe the drilling process in FRPCs and predict the levels of thrust force and torque that are considered to be the main contributing factors causing delamination. In a short span of time without the need for extensive experimental testing, such modeling enables us to optimize the drilling setup. It is desirable to apply the boundary conditions as closely as possible to the experiments. A reasonable compromise must be made between numerical accuracy and the computational choice, which depends upon the carefully selected numerical scheme, including mesh size and shape of elements. The major drawback of the FEM model in the case of drilling is the choice of an accurate friction model, which is still a field of ongoing research. Additionally, numerical models cannot simulate the effect of tool wear and lubrication on the results. Therefore, the experimental validation of FE models is often required to evaluate the accuracy of the proposed numerical methodology.

### 3.1. Material Modeling

Considering the fact that the ABAQUS material library is not equipped with a CDM-based model for 3D elements of FRPCs, a vectorized user material subroutine, abbreviated as VUMAT (Dassault Systemes Simulia Corporation, Providence, RI, USA), was used for the computation of the damage induced by the drilling operation. The GFRP sample plate was discretized with 3D solid elements to include the 3D state of stress, which is the actual scenario during the drilling. For establishing the interaction between the workpiece and drill bit, a surface-to-surface contact algorithm with the penalty method was employed. In addition, element deletion was also used during the analysis to exclude the complete damaged element from the computation to alleviate the computational cost. This also makes the model capable of predicting the hole-making process at different instants of time. Finally, the numerical results of thrust force and torque were compared with the experimental measurements.

### 3.2. Damage Initiation

For the intra-ply damage study, a ply-by-ply failure method was adopted where each glass ply was treated as a homogenized material with anisotropic elastic constants. In the case of UD ply, this scale is termed the meso-scale of the damage modeling [31,32]. Such damage modeling possesses a clear advantage over the homogenized laminate models, i.e., macro-models. Consequently, meso-scale modeling imparts freedom to users to introduce intra-ply as well as inter-ply damage using phenomenological models characterizing the complex interaction between them [3,32]. 

Intra-ply damage was initiated using Hashin’s and Puck’s failure theories developed for the failure of an individual ply of FRPCs [33,34]. The damage in the fibers was predicted using Hashin’s theory, whereas the matrix failure was predicted using Puck’s theory since it has been proved to be more accurate for matrix failures [35]. To delete the elements once the failure criterion was reached, the above-mentioned theories were implemented in the VUMAT subroutine using the ABAQUS/Explicit time step. Equations (1)–(3) for the initiation of tensile failure damage are shown below:

Hashin’s criteria for fiber damage in tension $\left(\sigma_{11}⩾0\right)$:(1)$${\left(\frac{\sigma_{11}}{S_{11}}\right)}^{2}+{\left(\frac{\sigma_{11}}{S_{12}}\right)}^{2}+{\left(\frac{\sigma_{11}}{S_{13}}\right)}^{2}=1,d_{ft}=1$$

Hashin’s criteria for fiber damage in compression (σ_11_ < 0):(2)$${\left(\frac{\sigma_{11}}{X_{1c}}\right)}^{2}=1,d_{fc}=1$$

Puck’s matrix damage initiation:(3)$$\begin{array}{c} \left[{\left(\frac{\sigma_{11}}{2X_{1t}}\right)}^{2}+\frac{\sigma_{22}^{2}}{\left|X_{2t}\cdot X_{2c}\right|}+{\left(\frac{\sigma_{12}}{S_{12}}\right)}^{2}\right]+\sigma_{22}\left(\frac{1}{X_{2t}}+\frac{1}{X_{2c}}\right)=1\\ \sigma_{22}+\sigma_{33}>0,d_{mt}=1\\ \sigma_{22}+\sigma_{33}>0,d_{mc}=1 \end{array}$$
where *σ*_11_, *σ*_22_, *σ*_33_ and *σ*_12_ are the components of the stress tensor in a single finite element. In addition, *d_ft_*, *d_fc_*, *d_mt_* and *d_mc_* denote the continuous damage variables evolving as a function of the external forces caused by drilling in fiber tension, fiber compression, matrix tension and matrix compression, respectively. Ply normal strengths are denoted by *X*_1*t*_, *X*_2*t*_ and *X*_2*c*_. Subscripts ‘1’, and ‘2’ describe the in-plane fiber direction and in-plane transverse directions, respectively, whereas subscripts ‘t’ and ‘c’ denote the tension and compression, respectively. Additionally, in-plane shear strengths are expressed by *S*_11_ and *S*_12_, whereas the out-of-plane shear strength is denoted by *S*_13_.

### 3.3. Damage Evolution

After the damage initiation, the material failure begins and a damage evolution criterion is required for the degradation in the stiffness caused by the load due to the drilling process. Initially, the material followed a 3D linearly elastic constitutive behavior prior to the damage based on the stiffness matrix of orthotropic material expressed by Equations (4) and (5):(4)$$\sigma =C_{0}\varepsilon$$
where ***C*_0_** is the undamaged elasticity matrix of the intact material, i.e., virgin material. The above Equation is expressed with Voigt notation where the stress (***σ***) and strain (***ε***) are treated as vectors as shown in Equation (5):(5)$$\left[\begin{array}{l} \sigma_{11}\\ \sigma_{22}\\ \sigma_{33}\\ \sigma_{12}\\ \sigma_{23}\\ \sigma_{31} \end{array}\right]=\left[\begin{array}{cccccc} C_{11}^{0} & C_{12}^{0} & C_{13}^{0} & 0 & 0 & 0\\ C_{12}^{0} & C_{22}^{0} & C_{23}^{0} & 0 & 0 & 0\\ C_{13}^{0} & C_{23}^{0} & C_{33}^{0} & 0 & 0 & 0\\ 0 & 0 & 0 & C_{44}^{0} & 0 & 0\\ 0 & 0 & 0 & 0 & C_{55}^{0} & 0\\ 0 & 0 & 0 & 0 & 0 & C_{66}^{0} \end{array}\right]\left[\begin{array}{c} \epsilon_{11}\\ \epsilon_{22}\\ \epsilon_{33}\\ \epsilon_{12}\\ \epsilon_{23}\\ \epsilon_{31} \end{array}\right]$$
where the entries in the undamaged stiffness matrix are computed with the following functions:(6)$$C_{11}^{0}=E_{1}\left(1-v_{23}v_{32}\right)\Delta$$
(7)$$C_{22}^{0}=E_{2}\left(1-v_{13}v_{31}\right)\Delta$$
(8)$$C_{33}^{0}=E_{3}\left(1-v_{12}v_{21}\right)\Delta$$
(9)$$C_{12}^{0}=E_{1}\left(v_{21}+v_{31}v_{23}\right)\Delta$$
(10)$$C_{23}^{0}=E_{2}\left(v_{32}+v_{12}v_{31}\right)\Delta$$
(11)$$\Delta =1/=\left(1-v_{12}v_{21}-v_{23}v_{32}-v_{13}v_{31}-2v_{12}v_{32}v_{13}\right)$$

Once the onset damage is initiated, the elastic matrix is changed from the above form to the following modified form to accommodate for the damage:(12)$$\sigma =C_{d}\varepsilon$$
where ***C_d_*** is the damaged elasticity matrix with individual components represented as:(13)$$C_{11}=\left(1-d_{f}\right)E_{1}\left(1-v_{23}v_{32}\right)\Delta$$
(14)$$C_{22}=\left(1-d_{f}\right)\left(1-d_{m}\right)E_{2}\left(1-v_{13}v_{31}\right)\Delta$$
(15)$$C_{33}=\left(1-d_{f}\right)\left(1-d_{m}\right)E_{3}\left(1-v_{12}v_{21}\right)\Delta$$
(16)$$C_{12}=\left(1-d_{f}\right)\left(1-d_{m}\right)E_{1}\left(v_{21}-v_{31}v_{23}\right)\Delta$$
(17)$$C_{23}=\left(1-d_{f}\right)\left(1-d_{m}\right)E_{2}\left(v_{32}-v_{12}v_{31}\right)\Delta$$
(18)$$C_{31}=\left(1-d_{f}\right)\left(1-d_{m}\right)E_{1}\left(v_{31}-v_{21}v_{32}\right)\Delta$$
(19)$$C_{44}=\left(1-d_{f}\right)\left(1-s_{mt}d_{mt}\right)E_{1}\left(1-s_{mc}d_{mc}\right)G_{12}$$
(20)$$C_{55}=\left(1-d_{f}\right)\left(1-s_{m}d_{mt}\right)E_{1}\left(1-s_{mc}d_{mc}\right)G_{23}$$
(21)$$C_{66}=\left(1-d_{f}\right)\left(1-s_{mt}d_{mt}\right)E_{1}\left(1-s_{mc}d_{mc}\right)G_{13}$$

Even though the damage variable can be treated as a continuous variable having any intermediate value between 0 and 1 depending on the magnitude of load in particular finite elements, it increases the computational cost significantly. As a consequence, in VUMAT, all the damage variables can attain a value of either 0 or 1, where 0 depicts the material as virgin material and 1 represents the particular element that has lost its complete load-carrying capability with no stiffness. Once the value of any failure index for a particular element is 1, the element is deleted from the computational domain. In other words, the composite is assumed to behave as a brittle material with no damage evolution.

## 4. Simulation Set-Up

### 4.1. Explicit Dynamics Modeling

In the present work, the 3D finite element set-up of the drilling process of GFRPs laminate was modeled in ABAQUS/Explicit. ABAQUS Explicit can handle highly non-linear behavior, such as contacts, effectively, making it a suitable choice. Unlike the ABAQUS/Standard where the accurate initial approximation is required, the explicit method incrementations are relatively inexpensive. The computational cost of the explicit dynamic simulation can be reduced either by reducing the number of increments required by speeding up the simulation as compared to the actual time of process or by artificially increasing density by a factor called mass-scaling. In our study, fixed mass-scaling was applied through the desired minimum stable time increment, for which the mass-scaling factors were determined by ABAQUS/Explicit. Several mass-scaling factors were studied and the time increment for which the total mass of the model did not change by more than one percent was used.

### 4.2. Geometrical Modeling and Boundary Conditions

To use 3D solid elements for material damage, the Vectorized User Material (VUMAT) subroutine was required in ABAQUS. Therefore, the material properties of the GFRP model were provided as mechanical constants under user material in the property module of the ABAQUS/CAE software. The material used in the current research was S2/FM94 lamina consisting of S2 glass fibers in the form of prepregs with individual ply thicknesses of 0.133 mm. Table 2 summarizes the mechanical properties of S2/FM94 material used in the drilling simulation. The workpiece consisted of a total of 54 layers with orientation as [0/90]_27s_.

The drill-bit modeled in the study was a 6 mm TiAlN-coated carbide drill-bit with a point angle of 140° and a helix angle of 30°. This typical drill bit is suitable for general use in FRPCs with an HRC range of 55–233. The drill was modeled using SolidWorks and imported into ABAQUS as an IGES file with discrete rigid attributes. The assumption behind modeling the drill bit as rigid is that the elastic stiffness of the tool is in the range of 500–700 GPa as compared to 54 GPa for the GFRP material in this study, as shown in Table 2. Thus, the tool was idealized as a rigid body in the computation that did not deform during the drilling process. It is emphasized that modeling the drill bit as a rigid body significantly reduced the computational resource during the simulation which is commonly followed in several studies [22,36].

### 4.3. Drill-Bit Work Piece Contact Definition

The contact and friction parameters considered in the current numerical study are based on values reported in the previous literature [3,26,27]. Accordingly, the contact between the twist drill-bit and all the plies of the laminate was defined as surface-to-node contact within the surface-to-surface contact algorithm available in ABAQUS/Explicit, whereas normal behavior was defined using hard contact. On the other hand, tangential behavior was defined using enforced penalty friction formulation without shear stress threshold and infinite inelastic stiffness. Furthermore, the coefficient of friction between the drill bit and the workpiece was defined as 0.2 based on the literature mentioned above. As far as the computational resources are concerned, the models require an average of 48 h utilizing 64 Intel quad-core processors with 356 GB RAM to complete the analysis. This high-performance computing (HPC) facility is available at the School of Interdisciplinary Engineering and Sciences (SINES) situated at the National University of Sciences and Technology (NUST), Islamabad, Pakistan.

### 4.4. Boundary Conditions

The workpiece was fixed at the four ends using the encastre function in ABAQUS. This function enforces zero translation and zero rotation to the workpiece at the pre-defined surfaces of interest. To apply the feed rate and cutting speed, a reference point (RP) was defined at the drill-bit tip. The drill bit was only allowed to move in the feed axis and was constrained in X- and Y-directions. Figure 1 shows the boundary conditions as applied in ABAQUS in addition to the RP.

### 4.5. Finite Element and Mesh Sensitivity

A mesh convergence study was carried out to ensure that the FEM results are independent of the element mesh size. Initially, a larger mesh size was used and the mesh size was reduced subsequently after each iteration until further reductions in mesh size did not affect the results. The mesh study was completed using the constant spindle speed of 3000 rpm at a feed rate of 300 mm/min. Based on the mesh convergence study, a final mesh of size 0.2 mm was used in the proximity of the hole, whereas a coarser mesh of 1.25 mm was used away from the drilling region (see Figure 2). The reason for this adaptive meshing was to reduce the computational time and achieve higher accuracy at the regions of interest.

A three-dimensional brick element, denoted by C3D8R, having eight nodes based on linear shape functions for the interpolation of the displacement in the workpiece with reduced integration, was selected from the ABAQUS element library. Only one element was modeled along the thickness direction of each ply. For meshing the drill-bit, rigid triangular-facet elements R3D3 were used with a constant element edge length of 0.25 mm. The total number of elements in the discretization was 110,430 in the workpiece and 10,048 rigid elements associated with the drill bit. The meshed configuration of the workpiece and drill bit is shown in Figure 2.

## 5. Results and Discussion

### 5.1. FEM Model Validation

Figure 3 shows the comparison between thrust force and torque profiles obtained from the numerical model and the experimental tests. It can be observed that the FE model slightly underpredicted the thrust force initially and then showed a good agreement with the experimental thrust force distribution. The initial underestimation is attributed to the simplified design of the drill bit. Apart from that, another reason might be related to the material model, which initially treated the material as brittle with no plasticity incorporated. In addition, the sudden damage mechanism implemented in VUMAT caused elements to fail, once the failure threshold was reached and the stiffness thus degraded instantly. Hence, this allowed the elements to fail prematurely, offering no resistance to penetration.

The average maximum thrust force in the FEM and experiments for 3000 rpm and 300 mm/min was 77 N and 80 N, respectively, which represented a 4% error. It may be noted that several factors can influence the accuracy of the numerical results, among them being the use of a more realistic friction model or friction coefficient. Additionally, the type of elements and the use of bulk viscosity coefficient can affect the thrust force and torque values.

The thrust force profiles show several stages, which are very well correlated with the tool position. For instance, in Figure 4, stage A started when the chisel edge made the first contact with the material and the corner of the cutting edge reached the first play of the material. Naturally, the thrust force is zero at the start and then increased rapidly as the chisel edge progressed into the material. This stage can be classified as the entry stage since the tool started to enter into the material. Next, stage B can be seen, which occurred when the chisel edge attained full contact with the workpiece material. Here, the thrust force reached a maximum value first and then a plateau can be noted when the drill bit was in full contact with the workpiece. Finally, stage C can be observed in Figure 4, which occurred when the dill bit was on the verge of exiting the workpiece from the bottom side. Then, the thrust force dropped until it reached zero, indicating the end of the drilling process.

### 5.2. Effects of Feed Rate and Spindle Speed

Several cutting parameters were selected to evaluate the effects of feed rate and spindle speed on the thrust force and torque. The comparison of the experimental and numerical findings is shown in Figure 5a for thrust force and Figure 5b for the corresponding torque. Since the thrust force can affect delamination, it can be stated that delamination decreases at higher spindle speeds. This observation is in line with the findings of previous experimental and numerical studies. The same can be observed from the state variable 3 denoted by SDV3 in Figure 6, which represents the matrix damage in compression, where it is higher at lower spindle speeds as compared to higher spindle speeds.

Table 3 shows the experimental and simulated thrust force in addition to torque values against the corresponding feed rate and spindle speed. The total error between the results can also be noted. This error can arise due to different reasons. The unavailability of an accurate friction model can result in errors between the predicted and measured values. Additionally, in the current study, no special attention was offered to modeling the accurate geometry of the drill bit, thus probably also contributing to the errors. On the other hand, the accuracy of the measuring sensors during the experiments can result in differences between FE and experimental results. Nevertheless, the overall average error is less than 10%, which is acceptable.

### 5.3. Effects of Friction Coefficient

The choice of friction coefficient (*µ*) and friction model is the key input data that are required when simulating the machining of FRPs. The identification of the correct coefficient of friction between FRPCs and the tool has still not been widely investigated. The main reason behind the lack of these data is the availability of relevant tribometers simulating the tribological conditions during machining. The majority of available tribometers are tribo-systems, such as pin-on-disc systems, where a pin is always in contact with the same part of the material. This situation is not the true representation of the cutting system, especially for FRPCs where the material is composed of several oriented layers [37].

Reductions in friction coefficient by using a lubricant during the machining of the laminated CFRPs are reported in [38]. The vast majority of previous research on composites drilling has considered the Coulomb Friction model with a constant coefficient [17,39,40,41]. In the present work, the Coulomb model was also utilized as model friction between the involved contact surfaces. The friction coefficient (*µ*) accounts for the shear stress from the surface traction (*τ*) with the contact pressure (*p*) and is expressed by Equation (22).
(22)τ=pµ

In most of the published literature, the coefficient of the friction value between the workpiece and tool varies in the range of 0.2–0.7 [42,43,44]. Therefore, the coefficient of friction in this range was investigated and compared with the experimental values of thrust force and torque for the cutting parameters of 3000 rpm and 300 mm/min, which were also used in the mesh convergence study earlier in Section 4.5.

Some studies have reported that the coefficient of friction has no significant effect on thrust force, and the thrust force in our study decreased with increasing friction coefficient. This is reasonably justified since, at a higher friction coefficient, the elements are deleted faster due to the high shear stress induced between the drill bit and the workpiece. The shear stress contributes to the damage presented in the prior mathematical formulations of Hashin’s and Puck’s failure theories in Section 3.2. However, modeling the exact relation between thrust force and friction coefficient can be a subject of future studies. 

Figure 7a shows the effect of the coefficient of friction on the fuzzing and spalling that occurred during the drilling of UD composites. Fuzzing is the result of the uncut fibers around the hole due to the acute angle between fiber and cutting velocity, whereas spalling is the delamination that develops further on the side of the drill. Figure 7b shows the thrust profiles of the cutting parameters of 3000 rpm and 300 mm/min for the 0.2 and 0.7 coefficients of friction, respectively.

## 6. Conclusions

In this paper, the effect of machining parameters on thrust force and torque along with the effect of friction coefficient when drilling S2-FM94 GFRPs was investigated numerically and validated with experimental results from the previous literature. A 3D FE drilling model was developed to account for 3D stress state damage. A user-defined material subroutine was used to model the orthotropic material response along with a stress-based damage criterion at the ply level. Hole drilling was visualized using an element deletion algorithm based on threshold stress levels for GFRPs using Hashin’s and Puck’s theories. The following observations were made during the analysis.

A more computationally inexpensive modeling approach was developed to predict the thrust and torque profiles. The FE model predicted the thrust force within a 10% error margin when compared with the experimental results. Thus, the current model can be used for future studies for more complex tool-workpiece interaction studies.The effect of feed rate and spindle speed on the thrust force was investigated. It was established that higher spindle speeds (7000 rpm) and lower feed rates (300 mm/min) are optimal for drilling. However, to compensate for machining time, lower feed rates can be utilized at the start and end of the drilling process and the rest of the process can be carried out using higher feed rates. Therefore, an investigation of variable feed rates is required.The effect of the friction coefficient was investigated. It was concluded that the prediction of the FE model is influenced greatly by the friction coefficient. A lower coefficient can increase the thrust force and a higher coefficient can reduce the thrust force. Further investigations on friction coefficient can be carried out in the future by utilizing different coefficients of friction according to the various materials of drill-bits.The current study did not examine the effect of back support on delamination because the analyzed workpiece was sufficiently thick. However, for thin workpieces (2–4 mm), a backing plate reduces failure at lower thrust loads. Therefore, for thin workpieces, it is essential to take into consideration the effect of backing plates in modeling.In this work, the selection of the drill bit was restricted by the experimental studies performed in [5]. However, tool geometry, especially the point angle and the geometry of the chisel edge, affect delamination. This can be subject to future studies along with other machining parameters.Research on cryogenic drilling of composites will be conducted in the future for its advantages/disadvantages over conventional drilling and an extension of our FE model presented in this paper will be utilized to simulate the damage of the same material under cryogenic conditions, which will be published in the near future.

## Figures and Tables

**Figure 1 materials-15-07052-f001:**
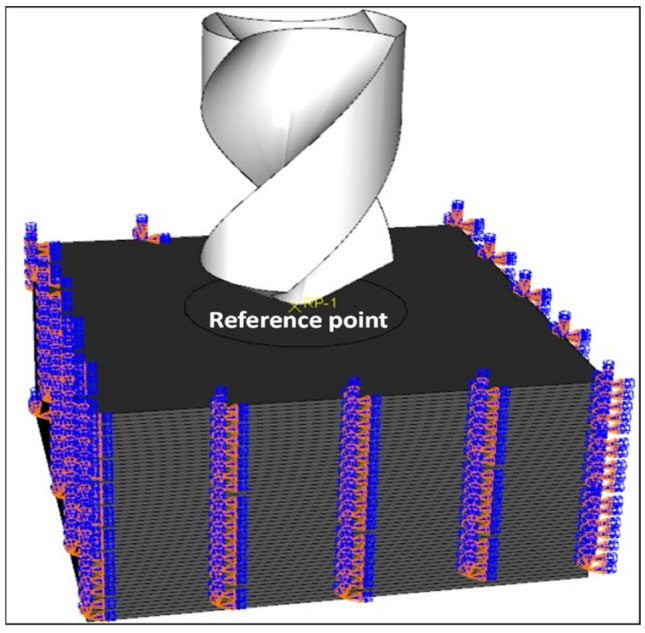
Drill-bit and workpiece along with the boundary conditions.

**Figure 2 materials-15-07052-f002:**
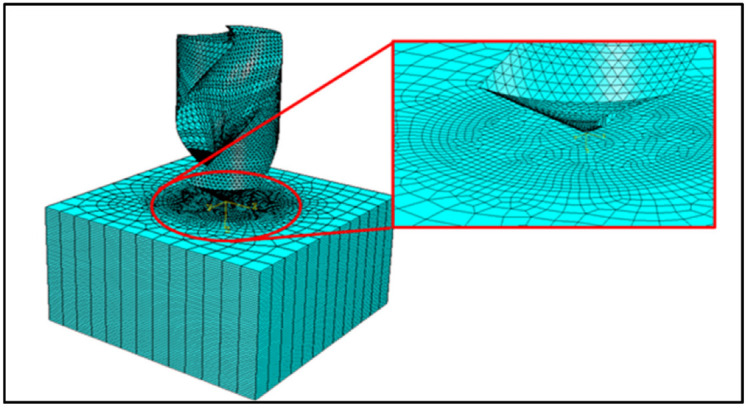
Finite element discretization of the domains.

**Figure 3 materials-15-07052-f003:**
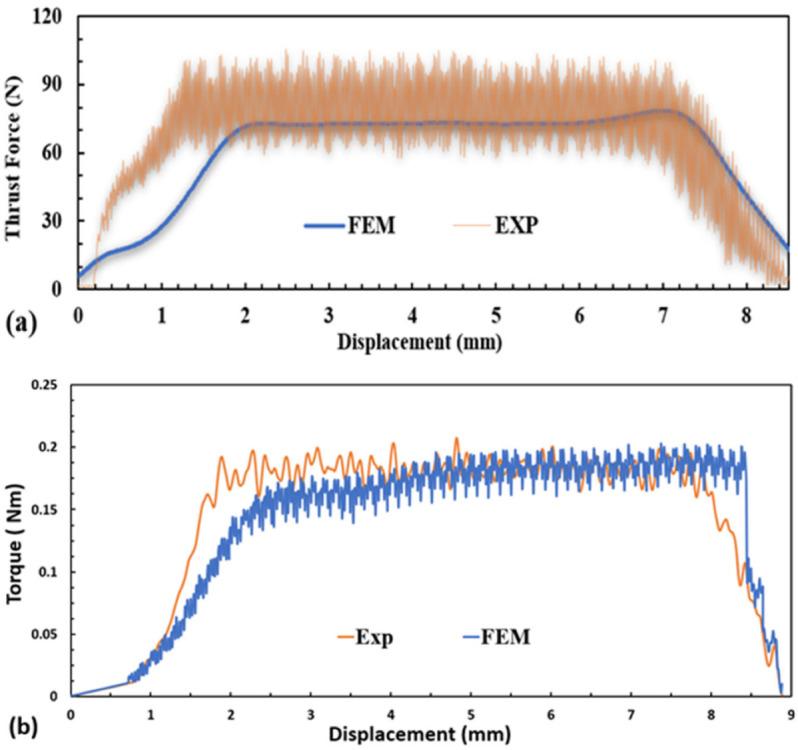
Comparison between FEM prediction and experimental results at a spindle speed of 3000 rpm and feed rate of 5 mm/s: (**a**) thrust force comparison and (**b**) torque profiles.

**Figure 4 materials-15-07052-f004:**
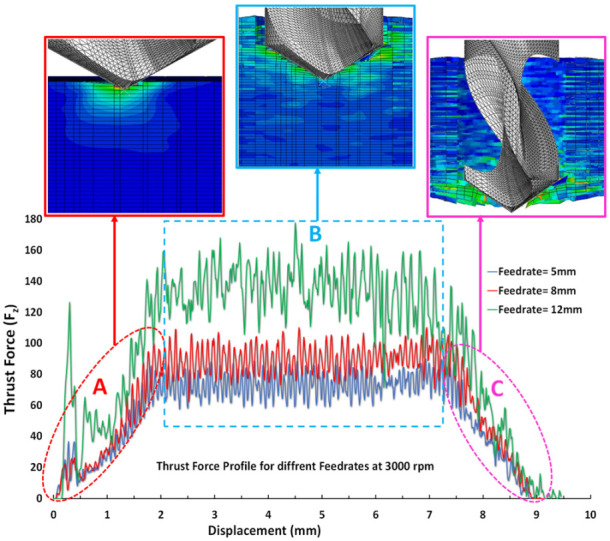
Different stages of thrust profiles during drilling.

**Figure 5 materials-15-07052-f005:**
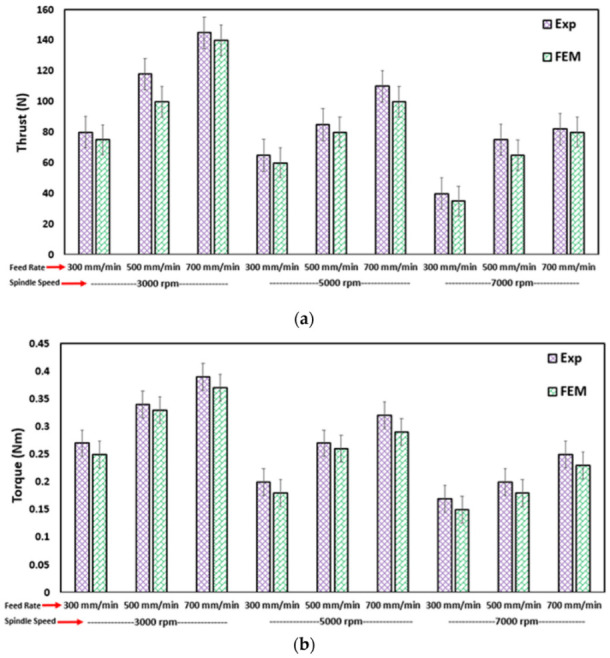
Comparison of the model prediction and experimental results at various feed rates and spindle speeds: (**a**) thrust force and (**b**) torque.

**Figure 6 materials-15-07052-f006:**
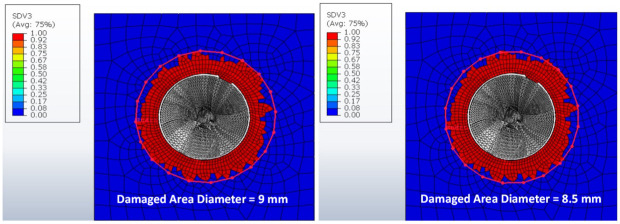
Illustration of drilling-induced damage by fuzzing and spalling: (**Right**) 3000 rpm and (**Left**) 5000 rpm.

**Figure 7 materials-15-07052-f007:**
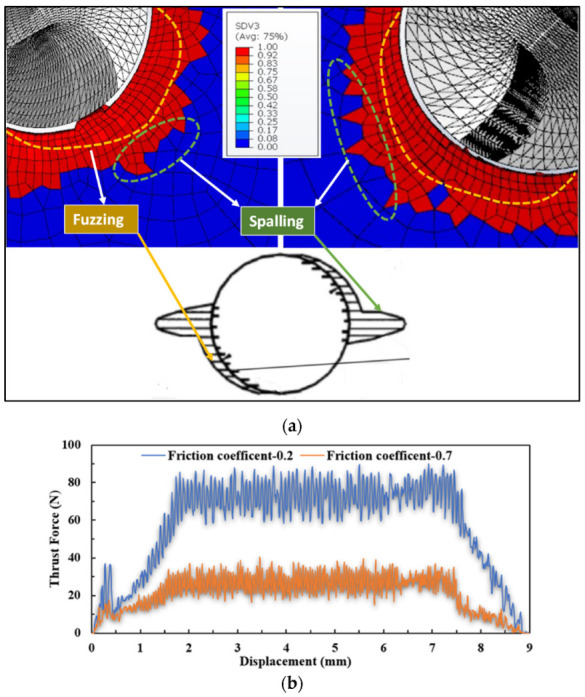
Comparison of (**a**) fuzzing and spalling damage (Right) *µ* = 0.7 (Left), *µ* = 0.2 (right) and (**b**) thrust profiles at different friction coefficients.

**Table 1 materials-15-07052-t001:** Experimental machining parameters [5].

Spindle Speed (rev/min)	Feed Rate 1 (mm/min)	Feed Rate 2 (mm/min)	Feed Rate 1 (mm/min)
3000	300	500	700
5000	300	500	700
7000	300	500	700

**Table 2 materials-15-07052-t002:** Homogenized individual ply material properties of S2/FM94.

Mechanical Property	UD S2/FM 94 Epoxy Prepreg		
Young modulus in 1-direction	*E_1_*	54-55	GPa
In-plane shear modulus	*G_12_*	5.55	GPa
Shear moduli	*G_13_ = G_23_*	3.0	GPa
Fiber direction tensile strength	*X_1T_*	2640	MPa
Transverse direction tensile strength	*X_2T_*	57.0	MPa
Poisson ratio	*v_12_*	0.33	*-*
Density	ρ	1980	Kg/m^3^

**Table 3 materials-15-07052-t003:** Experimental and FEM comparison of torque and thrust force.

Spindle Speed (Rpm)	Feed Rate (mm/min)	Exp. Thrust Force (N)	FEM Thrust Force (N)	Error %	Exp. Torque (N-m)	FEM Torque (N-m)	Error %
3000	300	83	75	9.6	0.27	0.25	7.4
	500	115	100	13.0	0.34	0.33	2.9
	700	150	145	3.3	0.39	0.37	5.1
5000	300	65	59	9.2	0.2	0.18	10.0
	500	85	75	11.8	0.27	0.26	3.7
	700	105	95	9.5	0.32	0.29	9.4
7000	300	45	30	33.3	0.17	0.15	11.8
	500	65	55	15.4	0.2	0.18	10.0
	700	83	77	7.2	0.25	0.23	8.0

## Data Availability

The cutting forces data and numerical results can be provided by contacting the corresponding authors.

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
