# Peer review of "Three-Dimensional Finite Element Modeling of Drilling-Induced Damage in S2/FM94 Glass-Fiber-Reinforced Polymers (GFRPs)"

_materials, 2022, doi:10.3390/ma15207052_

Round 1

Reviewer 1 Report (Previous Reviewer 2)

1. The innovativeness of your methodology must be presented. Emphasize what is new in methods and/or experimental research. Write the scientific contribution of your research and scientific hypotheses.

2. Furthermore, the authors note the following: "Unlike previous studies where a more computationally expensive methods like cohesive zone modelling were employed, the current model is numerically inexpensive." Such a claim must be supported with adequate references (citations). Such a claim must be supported by adequate references (citations). Is this a professional or scientific contribution of this research? Furthermore, how much inexpensive is it?

3. The results will naturally change when using other objects of the technological system or process system parameters. For example, when using a different type of machine, or the same type, but with a different, slightly modified stiffness; when machining a different material, or the same material, but with a different hardness, etc., etc. This makes it necessary to carry out a series of similar simulations and experiments each time. This study uses known known methods. Consequently, the presented work is engineering. How can you generalize about the research carried out and the results obtained?

4. In Figure 4, the position of the drill associated with detailed C is not clear. Why was the direction of the drill changed so much?

Author Response

Dear Reviewer,

Please find our response in the attachment! We have addressed all the points in the response.

Thanks!

Reviewer 2 Report (Previous Reviewer 1)

No comments at this stage. The authors have submitted the paper following the reviewer suggestion. The document should be accepted and considered for publication.

Author Response

Dear Reviewer,

We thank for the acceptance of our manuscript!

BR

Round 2

Reviewer 1 Report (Previous Reviewer 2)

The manuscript has been corrected.

This manuscript is a resubmission of an earlier submission. The following is a list of the peer review reports and author responses from that submission.

Round 1

Reviewer 1 Report

The abstract is concise and well written. The manuscript features an introduction, experimental parameters, finite element model of drilling, FEM model setup, results and discussion, and conclusions. The research methodology on problem definition is appropriate and properly applied. The paper is easy to read and free from grammatical or spelling errors, but some suggestions are recommended.

1-The ‘’Introduction’’ provides the necessary background information that covers the largest number of references but could be improved. It is suggested to cite and discuss further relevant recent literature related to numerical tests of drilling processes. Among others, authors can read some recommended references, as the following:

(2022). Proposal for a New Bioactive Kinetic Screw in an Implant, Using a Numerical Model. Applied Sciences; 12(2):779, DOI: 10.3390/app12020779.

2-If other dimensions (diameter, point and helix angle) will be used for the drill bit, what do you expect from the results?

3-The authors use Hashin and Puck’s theories to model damage initiation in the model. And about other models like Cowper-Symonds, Johnson-Cook, what is your opinion on this use?

4-About thermal evaluation, how could you know the influence of temperature on your model? I suggest that the authors expand their discussion considerably and perhaps provide a more scientific interpretation of different drilling parameters whether or not temperature actually influenced their model.

5-And about the drilling parameters, did the authors obtain optimum parameters such as drill speed, feed-rate and hole depth?

Reviewer 2 Report

The comments for improving the handwriting are as follows.

  1. The abstract needs to be improved. Clarify the objective and detail the research method. Include the study contribution. Abstract should contain some quantitative information also.
  2. Please correct the keywords. For example, "innovative tool design" does not appear anywhere in the manuscript.
  3. Further elaborate on the innovativeness of tool design.
  4. Introduction section is noticed to have limited literature survey about the research topic. There should be proper continuation on literature survey and how the research gap exists on this topic. Similarly, why this research topic is important?
  5. In the last paragraph of the Introduction section, the authors list the objectives of their research. However, the scientific contribution should be explicitly stated. The shortcomings of previous research should also be mentioned before that.
  6. Further elaborate on the selection of Machining Parameters shown in Table 1. It is not enough to refer to a reference. Why these parameters are representative of your research.
  7. Show detailed information about the cutting tool. Show all geometric parameters of the drill. Explain the choice.
  8. The universality of the methodology is not emphasized in the manuscript. What's new in methodology and / or numerical methods? The novelty is currently hidden. It only works as an application of FEM. Emphasize the novelty in the manuscript.
  9. Simulations also have drawbacks. I think they should at least be mentioned. Fine Element Methods has many advantages but also disadvantages.
  10. Experimental research would certainly be a good verification of the developed numerical model.
  11. Show the results in a table for all combinations of input variables. Show the results of the simulation and the results of the experiments in parallel.
  12. Two coefficients of friction are used. Are they static or kinematic values? Further elaborate the selection.
  13. Perform error estimation. Show in manuscript. Analyse and discuss errors.
  14. Further analyse and discuss the possibilities of practical application in industry.
  15. Do other elements of the machining system affect the results (fixtures, machine tools, lubrication, etc.). Further elaborate in the corrected manuscript.
  16. The article was submitted to the journal Materials. Therefore, the complete survey will have to contain detailed data from the materials of the cutting tool and the workpiece, the influence of drilling on the structure of materials, etc.

Round 2

Reviewer 2 Report

The manuscript has been significantly corrected. I recommend accepting the article in its current form.